# Analysis of the Prognostic Factors That Influence the Outcome of Periapical Surgery, including Biomimetic Membranes for Tissue Regeneration: A Review

**DOI:** 10.3390/biomimetics9050258

**Published:** 2024-04-24

**Authors:** Antonio J. Saiz-Pardo-Pinos, Francisco J. Manzano-Moreno, Esther Muñoz-Soto, María Paloma González-Rodríguez, Nuria Romero-Olid, María Victoria Olmedo-Gaya

**Affiliations:** 1Private Practice in Implant Dentistry, Avenida Andalucía, 69, 23005 Jaen, Spain; ajsaizpardo@gmail.com; 2Department of Stomatology, School of Dentistry, University of Granada, 18071 Granada, Spain; mpaloma@ugr.es (M.P.G.-R.); nromero@ugr.es (N.R.-O.); mvolmedo@ugr.es (M.V.O.-G.); 3Master of Oral Surgery and Implant Dentistry, School of Dentistry, University of Granada, 18071 Granada, Spain; msotodental@gmail.com

**Keywords:** periapical surgery, prognosis, guided tissue regeneration

## Abstract

The objective of this study was to analyze the prognostic factors that influence the outcome of periapical surgery. A systematic search of the literature was carried out using PubMed and Scopus databases between January 2000 and December 2023 with no language limitations. The PICO question of the present systematic review was: What prognostic factors may influence the outcome of periapical surgery? The most relevant randomized controlled clinical trials (RCTs), prospective clinical trials, retrospective studies, and meta-analyses (n = 44) were selected from 134 articles. The reviewed literature evidenced that bone-lesion healing could significantly be improved by the absence of deep periodontal pockets (>4 mm), localization in anterior teeth, the absence of pain and/or preoperative symptoms, a size of bone lesion < 5 mm, the use of ultrasound, the correct placement of retrograde filling material, and the use of different biomimetic membranes for guided tissue regeneration (GTR). Some preoperative and intraoperative factors could significantly improve the prognosis of periapical surgery. However, these results were not conclusive, and further high-quality research is required.

## 1. Introduction

Periapical surgery, also known as apicoectomy or root-end surgery, is a dental procedure performed to treat an infection or inflammation at the tip of the tooth root (apex) and the surrounding bone [1]. Periapical surgery is classically performed when orthograde endodontic treatment fails or when retreatment is not possible. The Spanish Society of Oral Surgery [2,3,4] proposed a series of indications for periapical surgery: periapical pathology in a tooth with a prosthodontic treatment that cannot be removed, periapical pathology of a permanent tooth that has a well-performed endodontic treatment with inflammation and pain, a radiolucent image greater than 8–10 mm in diameter, overfilling with gutta-percha, the presence of another foreign body that cannot be removed in an orthograde way, and other indications such as a fracture of the apical third of the root.

Clinical studies on periapical surgery outcomes have reported success rates ranging from 37% to 91% [5,6,7,8,9,10]. This wide variability may be due to differences in patient inclusion criteria, surgical approaches, magnification and illumination techniques, and obturation materials [7]. These differences make it difficult to conduct useful comparative studies or reviews [8], hampering the evidence-based evaluation by clinicians and patients of the risks, benefits, and costs of different treatment options. Knowledge of the likelihood of success of periapical treatment is important when deciding between this approach or the extraction of the tooth and its replacement with a fixed prosthesis or implant. This decision should combine optimal scientific evidence, clinical judgment, and patient preferences, taking account of the factors that influence treatment outcomes. However, limited data are available for only a few prognostic factors in periapical surgery, including age, sex, type of tooth, or presence/absence of a root-canal post [9,10,11,12,13,14,15]. There is a need for further analyses of presurgical factors (the distance and quality of root-canal fillings, restoration type, reason for surgery, tooth mobility, presence of fistulae, and periodontal status), surgical factors (the cavity preparation, retrograde root-end filling material, use of guided tissue regeneration, and experience of the surgeon), and post-surgical factors (crown sealing).

The aim of this systematic review was to explore and analyze all prognostic factors that might influence the outcome of periapical surgery divided among preoperative, intraoperative, and post-operative variables in order to help clinicians to increase the success of this type of treatment.

## 2. Materials and Methods

The PICO question of this systematic review was: What prognostic factors may influence the outcome of periapical surgery?

### 2.1. Search Strategy 

MEDLINE (using the PubMed search engine) and Scopus biomedical databases were used for a systematic electronic search of the literature between January 2000 and December 2023. The search strategy was (((periapical surgery) OR (apical surgery) OR (endodontic surgery) OR (apical microsurgery) OR (periradicular surgery) OR (apicoectomy) OR (apicoectomy) OR (root-end resection)) AND ((healing) OR (prognosis factors) OR (guided tissue regeneration) OR (biomimetic membranes)) NOT (case report OR case reports OR in vitro OR experimental)). The aim was to include all randomized clinical trials, prospective clinical trials, retrospective studies, and meta-analyses related to the prognostic factors potentially influencing the outcome of periapical surgery. We used algorithms and search strategies that could be reproduced by any researcher. Before starting the review, the protocol was registered in PROSPERO with the registration number ID535642.

### 2.2. Study Selection Criteria

The inclusion criteria were randomized clinical trials (RCTs), clinical trials, retrospective studies, meta-analyses, and studies using humans. The exclusion criteria were failures to address the PICO question, periapical surgery without the placement of retrograde filling materials, experimental studies, or case report designs.

### 2.3. Review and Screening of Articles

The titles and abstracts of the different studies collected using all search methods were independently assessed by two reviewers to determine compliance with the eligibility criteria. Differences of opinion between the reviewers were resolved by discussions between the two reviewers and if there was no consensus, a third reviewer was consulted. An analysis of the results was carried out to eliminate duplication. The level of agreement between the reviewers on the inclusion of studies was expressed using the kappa index. All studies that met the inclusion criteria were included and data extraction was performed. Reasons for excluding articles from the review were recorded and discussed.

## 3. Results and Discussion

A total of 1209 articles were found: 639 in PubMed and 570 in Scopus. The search strategy and results are shown in detail in Figure 1. After eliminating duplicate articles (753), those without full text (62), and those not within the study objective (197), a total of 197 articles were selected for an in-depth analysis against the inclusion and exclusion criteria (99 via PubMed and 98 via Scopus). Finally, 44 articles were included in the review (37 via PubMed and 7 via Scopus), comprising 17 RCTs, 15 prospective clinical trials, 7 retrospective studies, and 5 meta-analyses (Table 1). The kappa value obtained was 0.91. 

### 3.1. Preoperative Factors

#### 3.1.1. Sex

Numerous authors have evaluated the influence of sex on the outcome of periapical surgery. The immense majority of studies found no statistically significant relationship between sex and treatment success [7,12,13,14,15,17,19,20], although outcomes for female patients were reported to be significantly better by Song et al. [15] and non-significantly better (*p* = 0.09) by von Arx et al. [12]. Superior outcomes were reported to be non-significant in males than in females by Tsesis et al. [34], while Taschieri et al. and Martí-Bowen et al. [24,35] observed significantly better results in males after six months but not after one year. 

#### 3.1.2. Age

Most of the relevant studies found no association between the age of the patient and the outcome of periapical surgery [6,19,21,36]. However, some authors reported earlier and improved healing in younger versus older patients [35,37]. Song et al. observed significantly superior outcomes for patients under 20 years old [15]. In contrast, Wang et al. [21] and Barone et al. [33] observed better outcomes for patients over 45 years old; these findings support periapical surgery as a predictable treatment option for older patients, with no apparent impairment in periapical tissue healing after adequate apical sealing using retrograde filling materials. 

#### 3.1.3. Periodontal Status

Periodontal status is widely considered to be a key prognostic factor and a good and stable periodontal status is considered to be a prerequisite for periapical treatment [6,22,24,35,38]. Wang et al. [21] demonstrated that preoperative marginal bone loss (periodontal probing depth > 4 mm) has a negative impact on the success of endodontic surgery (*p* < 0.03). A poor periodontal status was an exclusion criterion in some of the reviewed studies. Gagliani et al. [22] excluded patients with a probing depth > 6 mm and Zuolo et al. [6] excluded those with a probing depth > 7 mm in their longitudinal studies (5 years and 4 years, respectively). In another five-year study, Wesson and Gale [19] found that the success rate significantly decreased with increased marginal bone loss. In a prospective clinical study on endodontic microsurgery, Kim et al. [29] observed a success rate of 95.2% in patients with endodontic lesions alone versus 77.5% in those with endoperiodontal lesions, suggesting that this type of combined lesion has an adverse effect on soft tissue and bone healing. In summary, the periodontal pocket depth is widely accepted as an important prognostic factor in periapical surgery.

#### 3.1.4. Type of Tooth

Twenty-one studies provided information on comparative healing rates for different types of teeth, classified as anterior or premolar teeth or maxillary or mandible molars [4,5,6,11,12,13,14,18,19,22,23,24,25,26,27,31,37,39,40]. Superior success rates have been reported for anterior teeth than in premolars or molars, which may be explained by the easier surgical access and their less complex root anatomy. Wälivaara et al. [39] applied ultrasound and retrograde IRM^®^ root-end fillings to 56 teeth and obtained success rates of 100% for incisors, 78% for molars, and 69% for premolars. von Arx et al. [18], who used Retroplast^®^, found that the majority of failures were in premolars or molars, while Garcia et al. [31] reported a success rate of only 75% in premolars and molars at one year post-surgery. Wesson and Gale [19] observed a significant difference in success rates between mandibular first (60%) and second (46%) molars, which could be attributable to the thicker cortical bone in the posterior versus anterior mandible and the need to take account of the mentonian nerve or inferior dental nerve. Furthermore, the cutting angle of the root apex must sometimes be increased to improve its visibility in mandibular molars, augmenting the number of exposed dentinal tubules and the consequent microfiltration. 

#### 3.1.5. Preoperative Pain or Symptomatology 

The outcome of periapical surgery is also influenced by preoperative pain or clinical signs such as inflammation or the presence of fistulae. In their study, von Arx et al. [12] found initial pain to be the only significant prognostic factor. In a subsequent meta-analysis, they confirmed that a successful outcome was significantly (*p* < 0.01) more likely in patients with than without preoperative pain or symptoms. The reasons for this difference are poorly understood, although it has been speculated that the pain and/or symptoms may be associated with a sub-acute infection that can compromise surgical wound-healing [13].

#### 3.1.6. Endodontic Status

Endodontic status takes into account the distance and quality of the filling. Jensen et al. [10] and von Arx et al. [12] found that the preoperative endodontic distance had no significance influence on the final outcome, whereas Platt and Wannfors [20] evidenced a higher success rate when it was correct. Wesson and Gale [19] also observed a correlation between periapical surgery success and appropriate canal treatment. By contrast, Wang et al. [21] reported a success rate of 85% when the endodontic filling was inadequate and only 65% when it was adequate. Barone et al. [33]—and more recently, Song et al. [15]—also obtained significantly (*p* = 0.02) superior outcomes when the filling did not reach the end of the canal. Lustmann et al. [9] attributed these findings to the surgical removal of the unfilled end of the apex, considered to be the main focus of infection. The density of the filling was also described as a significant prognostic factor (*p* < 0.01) by von Arx et al. [13]. At any rate, the success of periapical surgery is more likely if the endodontic treatment is conducted as correctly as possible. When endodontic treatment fails, the first option is always to repeat the procedure and periapical surgery is only conducted when this is not possible or fails again [4].

#### 3.1.7. Presence of Root-Canal Post

Jensen et al. [10] and von Arx et al. [13] found no significant correlation between the healing rate and the presence or absence of a post or screw. However, a significantly (*p* = 0.051) higher rate was observed by Maddalone and Gagliani [17] in teeth with no posts (>97%) than in teeth with posts (88%), and Rahbaran et al. [5] also reported a significant difference in the same direction (*p* = 0.047), likely due to the presence of root fractures or cracks in teeth with posts. It is, therefore, important to use all available advances and intraoperative diagnostic technologies to detect these defects in teeth with posts. From a clinical perspective, the length of the post is more important than its presence or absence in periapical surgery. Given that current surgical recommendations include an apical resection of 3 mm and a retrograde cavity preparation to a further depth of 3 mm, a long post may exceed these distances and, hence, compromise the surgery and its outcomes. 

#### 3.1.8. Lesion Size

Most studies found no statistically significant relationship between lesion size and treatment success [10,11,27,29,31,32,37], although some considered it to be a clear prognostic factor [4,13,21,33]. Wang et al. [21] reported a superior prognosis in preoperative lesions ≤ 5 mm than in those >5 mm (*p* = 0.023). von Arx et al. [12] also observed improved periapical surgery outcomes in teeth with no preoperative radiologic lesions (94.1%) than in those with small (<5 mm) or large (>5 mm) lesions (86.5% and 77.1%, respectively), although a statistical significance was not quite reached (*p* = 0.06). Peñarrocha et al. [4] studied the relationship between the radiographic periapical lesion size, apical resection, and retrograde filling and the prognosis of periapical surgery, concluding that outcomes were improved with a smaller periapical lesion size and smaller resection and did not depend on the amount of the retrograde filling. They suggested that when the initial size is small, the pathological lesion is eliminated, whereas curettage may be incomplete in larger lesions due to anatomical impediments and the residual pathological tissue may become a bacterial reservoir for recurrent infections. Taschieri et al. [28] reported significantly superior outcomes in lesions with the loss of vestibular and palatal cortical bone after guided tissue regeneration (GTR) using bovine inorganic bone and resorbable membranes than in those not receiving this treatment (success rate 88.2% versus 57.1% in controls). Barone et al. [33] reported that a bone defect < 10 mm is a significant favorable prognostic factor, while von Arx et al. [13] found that the absence of a lesion or a lesion size < 5 mm significantly increased post-operative healing rates.

### 3.2. Intraoperative or Treatment-Related Factors 

#### 3.2.1. Type of Surgery: First Surgery versus Repeat Surgery

Although some authors [11,22,30] found significantly higher healing rates after the first periapical surgery in comparison with repeat surgery, no significant difference was observed in most studies [12,15,18,21,33]. In contrast, Rahbaran et al. [5] observed a lower rate after first (25.4%) versus repeat (34%) surgeries, although these results should be interpreted with caution because current techniques were not available to the authors 10 years ago. With modern surgical techniques, including the use of 4.5× magnifying glasses and ultrasound, Gagliani et al. [22] observed significantly superior success rates after first (86%) versus repeat (59%) surgeries. Saunders [30] obtained a 74.5% success rate after repeat surgery using MTA^®^ as a retrograde root-end filling material and modern surgical techniques, while a success rate of 92.9% for repeat surgery was recently reported by Song et al. [15], who used MTA^®^ and SuperEBA^®^ as retrograde root-end filling materials. They reported that the most common causes of failure for first surgery were the absence of retrograde filling material in part of the canal (44.4%) and incorrect retrograde cavity preparation (37%).

#### 3.2.2. Retrograde Cavity Preparation 

Higher success rates have been reported with the use of ultrasound for retrograde cavity preparation in comparison with rotary instruments [24,26,32]. In a double-blind randomized clinical trial (n = 290 patients), de Lange et al. [26] obtained a healing rate of 80.5% when ultrasound was employed for cavity preparation versus 70.9% when drilling was used and the difference was statistically significant for molars. In another double-blind randomized clinical study, Christiansen et al. [32] reported a success rate of 96% in a group of 18 teeth that underwent cavity preparation and were filled with MTA^®^ versus only 52% in a group of 18 teeth with no cavity preparation and filled using the cold-burnished gutta-percha method. Most authors [4,13,21,24,26,35,37,39,40] currently recommend the use of ultrasound for retrograde cavity preparation.

#### 3.2.3. Retrograde Root-End Filling Material 

The selection of retrograde root-end filling material is less clear than the choice of cavity preparation method because of the influence of confounding variables in clinical studies. Some authors consider that the choice of filling material is less important than the selection of cases (indication) or the technique used to place the material [38], among other factors. In general, the majority of materials that yielded good results in vitro achieved good success rates in patients. Thus, satisfactory outcomes have generally been reported for EBA [15,17,24], IRM^®^ [6,23,26,27,37,38,39,40], composites [10,12,18,41], and compomers [20]. Some studies also reported high success rates for silver amalgam [4,35], although others described worse rates in comparison with other materials [5,6,33]. The more recent MTA^®^ has not been evaluated as extensively as the other materials, but the clinical results have been very good [12,15,18,23,29,30,32,33,38] and it appears to have an encouraging future, given its exceptional sealing capacity and its biocompatibility in vitro and in vivo [18]. MTA^®^ has consistently demonstrated success rates > 90%. Two double-blind randomized clinical trials showed no statistically significant differences between MTA^®^ and IRM^®^ [23,38], while Wälivaara et al. [27] recently reported a success rate of 91% with IRM^®^ versus 82% with Super-EBA^®^, but the difference did not reach significance. Among the most widely studied materials, the worst clinical outcomes have been reported for glass ionomer, probably due to its high sensitivity to humidity [20].

#### 3.2.4. Experience of the Surgeon 

Some authors found no significant difference in success rates between highly experienced professionals and post-degree students [11,15], although Rahbaran et al. [5] reported worse outcomes of endodontic surgery with operators who had less experience. More recently, Chong and Ford [16] argued that this procedure is not appropriate for inexpert operators and should be carried out by hospital dentists, although they also emphasized that good surgical skills alone are not sufficient and that correct case selection and knowledge of the biological bases of the treatment are also needed. Surprisingly, Wang et al. [11] reported that the prognosis was superior for patients treated by post-degree students versus experienced clinicians, although the authors acknowledged that clinicians were given the most difficult cases (molars and premolars) and those with a worse initial prognosis.

#### 3.2.5. Guided Tissue Regeneration (GTR)

Several regeneration techniques and materials have been proposed for the healing of bone defects after surgical endodontic treatment (Table 2) [42,43,44,45,46,47,48,49,50,51,52]. Some authors found statistically significant differences between the use of GTR using biomimetic membranes and conventional techniques (without membranes). Better results were achieved when GTR was applied [42,43,44,45]. In this sense, Tobon et al. [42] found better radiographic results in terms of bone healing when a non-absorbable membrane (Goretex) + hydroxylapatite (Osteogen) were used when compared with a conventional technique (without a membrane) (P.0.016). Dominiak et al. [43] also found significant differences (*p* = 0.0408) between the control group and GTR groups (with resorbable collagen membrane + xenograft), with better bone healing in the experimental groups. Other authors of different systematic reviews and meta-analyses showed that GTR techniques improved periapical lesion healing after endodontic surgery [43,44]. In contrast, other authors found no significant differences in terms of bone healing after periapical surgery using different biomimetic membranes compared with conventional techniques (without membranes) [47,48,49,50].

#### 3.2.6. New Technologies 

Systems are being developed to amplify and increase the illumination of the surgical field. The utilization of magnifying glasses, surgical microscopes, and/or endoscopes, among other new systems, facilitates the work with instruments and retrograde cavity filling [13,25,30,37]. In a meta-analysis of 38 studies by von Arx et al. [13], a superior healing rate was found when an endoscope was used but this was found to be a significant prognostic factor in only two studies [12,41]. In contrast, Taschieri et al. [25] and Tsesis et al. [14] found no significant differences as a function of the type of magnification in their study. 

### 3.3. Post-Operative Factors 

#### Crown-Sealing Evaluation

Crown sealing has been described by various authors as one of the most important factors for success in endodontic treatment and periapical surgery. Rahbaran et al. [5] reported that complete healing was three-fold more likely in teeth with good crown restoration than in those with no restoration. An adequate and long-lasting crown seal prevents entry into the canal system of residues or products from the oral cavity that could otherwise compromise initial post-surgical wound-healing and the success of treatment. However, Barone et al. [33], Song et al. [15], and other authors [11,21,32,37] found that the type of tooth restoration did not significantly influence the long-term outcomes.

## 4. Conclusions

Most of the reviewed literature evidenced that the previous absence of deep periodontal pockets (>4 mm), localization in anterior teeth, the absence of preoperative pain and/or symptoms, a bone-lesion size < 5 mm, the use of ultrasound, and a correct retrograde root-end filling could significantly improve the prognosis of periapical surgery. However, contradictory results have been published on some aspects, including the influence of the periodontal status, endodontic status, presence of a root-canal post, type of root-end filling material, or the use of different biomimetic membranes for guided tissue regeneration (GTR). Further high-quality scientific research is required in order to obtain definitive conclusions.

## Figures and Tables

**Figure 1 biomimetics-09-00258-f001:**
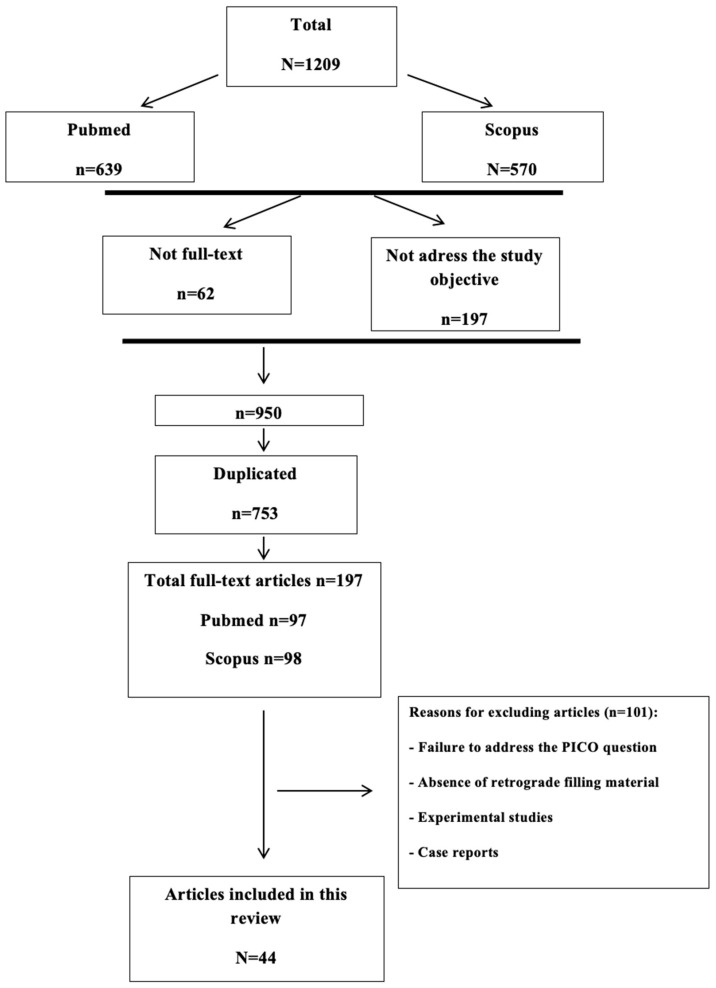
Flow diagram of literature search.

**Table 1 biomimetics-09-00258-t001:** Studies assessing the prognostic factors associated with periapical surgery success included in the review (N = 33).

	Zuolo 2000 [6]	Rahbaran 2001 [5]	Jensen 2002 [10]	Chong 2003 [16]	Maddalone 2003 [17]	von Arx 2003 [18]	Wesson 2003 [19]	Platt 2004 [20]	Wang 2004 [11]	Wang 2004 [21]	Gagliani 2005 [22]	Lindeboom 2005 [23]	Taschieri 2005 [24]	Taschieri 2006 [25]	Tsesis 2006 [14]	de Lange 2007 [26]
Study design	RS	RS	RCT	RCT	PS	PS	PS	RCT	PS	RS	PS	RCT	RCT	RCT	RS	RCT
Initial sample	114	314	134	131	146	129	1007	34	155	238	185	100	50	80	110	399
Final sample	102	176	122	108	120	115	790	34	90	194	164	100	46	71	71	290
Follow-up time (years)	1–4	4	1	2	3	1	5	1	4–8	1	5	1	1	1	1–4	1
Success rate (%)	91 ^ϕ^	37 ^θ^ 19 ^ғ^	73 ^c^ 37 ^з^	92 ^ɱ^ 87 ^ϕ^	92 ^θ^	88 ^c^ 75 ^c^	57 ^ғ^	89 ^℧^ 44 ^з^	74 ^θ^	…	78 ^θ^	92 ^ɱ^ 86 ^ϕ^	91 ^θ^	92 ^θ^	91 ^ϕ^	80 ^ϕ^ 70 ^ϕ^
Sex				…	…	…		…	…		…	…	…	…		…
Age				…	…	…		…			…	…	…	…		…
Periodontal status	…	…		…	…	…			…		…	…	…	…	…	…
Type of teeth			…	…	…	…		…	…							
Lesion type	…	…	…	…	…	…	…		…	…	…	…	…	…	…	…
Previous pain and symptoms	…			…	…	…	…	…	…		…	…	…	…		…
Post-surgical pain and symptoms	…	…		…	…	…	…	…	…	…	…	…	…	…	…	…
Endodontic status	…			…	…	…					…	…	…	…		…
Post	…			…	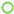	…	…	…	…	…	…	…				…
Lesion size	…			…	…	…	…	…			…	…	…	…		…
Type of surgery	…		…	…	…	…	…	…				…	…	…	…	…
Retrograde cavity preparation	…		…	…	…	…	…	…	…		…	…		…		
Filling material	…				…	…	…		…		…		…	…	…	…
Surgeon’s experience	…		…	…	…	…	…	…	…		…	…	…	…	…	…
New technologies	…	…	…	…	…		…	…	…	…	…	…	…			…
Crown sealing	…		…	…	…	…	…	…			…	…	…	…		…
	**Wälivaara 2007** [27]	**von Arx 2007** [12]	**Taschieri 2007** [28]	**Peñarrocha 2007** [4]	**Kim 2008** [29]	**Saunders 2008** [30]	**Taschieri 2008** [25]	**García 2008** [31]	**Christiansen 2009** [32]	**Tsesis 2009** [14]	**Wälivaara 2009** [27]	**Barone 2010** [33]	**von Arx 2010** [18]	**von Arx 2010** [13]	**Song 2011** [15]	**Song 2011** [15]	**Wälivaara 2011** [27]
Study design	PS	PS	PS	PS	RCT	PS	RCT	PS	RCT	MET	RCT	PS	PS	MET	RS	PS	RCT
Initial sample Final sample	56 55	194 191	30 27	363 363	263 188	321 276	34 31	97 92	42 36	11*	160 147	261 134	353 339	38*	907 491	54 42	206 194
Follow-up time (years)	1	1	1	1	2	1	1	1	1	….	1	4–8	1	….	1	2	1
Success rate (%)	80 ^ϕ^	83 ^ɱ^ ^θ c^	93 ^θ^	74 ^ғ^	91 ^ɱ^ 77 ^θ^	88 ^ɱ^	88 ^θ^ 57 ^θ^	75 ^ғ^	96 ^ɱ^ 52 ^g^	91	85 ^ϕ^ 90 ^g^	74 ^ɱ^ ^F θ^	91 ^ɱ^ 79 ^c^	…	…	92 ^ɱ^ ^θ^	91 ^ϕ^ 82 ^θ^
Sex	…		…	…	…	…	…	…	…		…	…		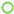		…	…
Age	…		…	…	…	…	…	…	…		…					…	…
Periodontal status	…	…	…	…		…	…	…		…	…	…	…	…	…	…	…
Type of teeth					…	…	…		…			…				…	
Lesion type	…	…	…	…		…		…		…	…	…	…	…		…	
Previous pain and symptoms	…		…	…	…	…	…	…		…	…	…	…			…	…
Post-surgical pain and symptoms	…	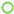	…	…	…	…	…		…	…	…	…	…	…	…	…	…
Endodontic status	…		…		…	…	…	…		…	…		…			…	
Post	…			…	…		…	…	…	…	…	…				…	…
Lesion size	…	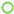	…			…	…			…	…		…		...	…	
Type of surgery	…		…	…	…		…	…	…	…	…			…		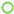	…
Retrograde cavity preparation	…	…	…		…	…	…	…		…	…		…	…	…	…	…
Filling material	…	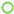	…	…	…	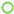	…	…				…		…		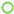	
Surgeon’s experience	…	…	…	…	…	…	…	…	…	…	…	…	…	…		…	…
New technologies	…	…		…	…	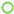		…	…		…	…	…		…	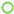	…
Crown sealing	…	…	…	…	…	…	…	…		…	…		…	…		…	…

…: Not assessed; 

: statistically significant (*p* < 0.05); 

: not statistically significant (*p* > 0.05); 
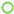
: close to significant association. **RCT**: Randomized controlled clinical trial; **PS**: prospective study; **RS**: retrospective study; **MET**: meta-analysis (11* and 38* studies included). Retrograde root-end filling material used: ^ɱ^: MTA; ^θ^: Super-EBA; ^ϕ^: IRM; ^c^: composite; ^з^: ionomer; ^ғ^: amalgam; ^℧^: compomer; ^g^: gutta-percha.

**Table 2 biomimetics-09-00258-t002:** The effect of guided tissue regeneration (GTR) in the prognosis of periapical surgery (N = 11).

Author	Study Design	Initial and Final Sample	Follow-Up Time (Months)	Type of Treatment	Guided Tissue Regeneration (GTR) versus Control
Tobón 2004 [42]	PS	28 28	12	1. Conventional. 2. Non-absorbable membrane (Goretex). 3. Non-absorbable membrane (Goretex) + hydroxylapatite (Osteogen).	 (Only between groups 1 and 3)
Marín-Botero 2006 [46]	RCT	30 30	12	1. Periosteal graft. 2. Bioabsorbable membrane of poliglactin 910.	
Dominiak 2009 [43]	PS	106 106	6–12	1. Control group. 2. Resorbable collagen membrane. 3. Xenogenic bovine material. 4. Xenogenic bovine material + Platelet-rich plasma.	
Tsesis 2011 [47]	MET	11*	…	…	
Taschieri 2011 [48]	RS	40 33	48	1. Guided tissue regeneration + xenogenic bone graft.	
Parmar 2019 [49]	RCT	40 32	12	1. Control (without membrane). 2. Collagen membrane (Healiguide).	
Liu 2020 [44]	MET	11*	…	1. Control (without membrane). 2. Non-absorbable membrane (e-PTFE).3. Collagen membrane. 4. Collagen membrane + bovine-derived hydroxyapatite. 5. Autologous platelet concentrates.	 (Only between groups 1 and 4)
Zubizarreta-Macho 2022 [45]	MET	11*	…	1. Control. 2. Bone graft. 3. Platelet-enriched plasma. 4. Membrane. 5. Membrane + bone graft. 6. Membrane + platelet-enriched plasma.	
Johri 2022 [50]	RCT	34	6	1. Amniotic membrane. 2. Platelet-rich fibrin (PRF).	
Garg 2023 [51]	RCT	19	12	1. Platelet-rich fibrin (PRF). 2. Mineralized freeze-dried bone allograft (FDBA).	
Albagle 2023 [52]	RCT	86	12	1. Control (without membrane). 2. Resorbable collagen-based bone-filling material.	

…: Not assessed; 

: statistically significant (*p* < 0.05); 

: not statistically significant (*p* > 0.05); **RCT**: randomized controlled clinical trial; **PS**: prospective study; **RS**: retrospective study; **MET**: meta-analysis (11* studies analyzed in each meta-analysis).

## Data Availability

The data presented in this study are available on request from the corresponding author.

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
