# Peer review of "Analysis of the Prognostic Factors That Influence the Outcome of Periapical Surgery, including Biomimetic Membranes for Tissue Regeneration: A Review"

_biomimetics, 2024, doi:10.3390/biomimetics9050258_

Round 1
Reviewer 1 Report
Comments and Suggestions for Authors
The manuscript by Saiz-Pardo-Pinos et al. is highly informative, well-organized, and valuable for the scientific literature. However, some simple suggestions for its improvement are listed below:
1. The "Introduction" section is better to begin introducing the periapical surgery as a method. What are the indications and contraindications for its application?
2. The aim of the study should contain some sentences summarizing the contributions and values of this research to the scientific literature.
3. Song et al. [11] (line 134).
4. A figure summarizing the results should be added for better understanding and statistical interpretation.
5. The overall number of references can be increased by adding more sources in the "Introduction" section and as a part of the discussion.
Author Response
Responses to #reviewer 1:
- Concern of the reviewer: The "Introduction" section is better to begin introducing the periapical surgery as a method. What are the indications and contraindications for its application?
Our response: Thank you for your question. We have modify the introduction section according to your suggestions and we have added the indications and contraindications for its application.
Revised text: Lines 32-43: “Periapical surgery, also known as apicoectomy or root-end surgery, is a dental procedure performed to treat an infection or inflammation at the tip of the tooth root (apex) and the surrounding bone [1]. This condition typically occurs when a root canal treatment fails to resolve the issue or when retreatment is not feasible. The indications for periapical surgery, based on the protocol proposed by the Spanish Society of Oral Surgery [2–4] are: 1) periapical disease affecting a permanent tooth subjected to endodontic treatment (of good quality), with pain and inflammation; 2) periapical pathology with prosthodontic or conservative restoration proven to be difficult to remove; 3) a radiotransparent lesion measuring over 8 to 10 mm in diameter; 4) symptomatic gutta-percha overfilling, or presence of a foreign body not amenable to orthograde removal (eg, fractured file); 5) other indications (patient requiring endodontic treatment and periapical surgery in a single session, fracture of the apical third, etc.).”
- Concern of the reviewer: The aim of the study should contain some sentences summarizing the contributions and values of this research to the scientific literature.
Our response: We have modified the aim of the study according to your suggestion.
Revised text: Lines 61-64: “The aim of this systematic review was to explore and to analyze all prognostic factors that might influence the outcome of periapical surgery, divided among preoperative, intraoperative, and postoperative variables in order to help the clinician to increase the success of this type of treatment.”
- Concern of the reviewer: Song et al. [11] (line 134).
Our response: We have corrected the mistake.
Revised text: Line 145: “…Song et al.[15] …”
- Concern of the reviewer: A figure summarizing the results should be added for better understanding and statistical interpretation.
Our response: Thank you for your suggestion. Due to the large number of articles included and because this is a review of the literature and not a meta-analysis, we have not performed statistics, therefore, it is difficult to make a figure that summarises all the findings, so we have reflected the more detailed results in two tables. However, figure 1 is attached, which summarises all the search and selection of articles.
Revised text: Not applicable.
- Concern of the reviewer: The overall number of references can be increased by adding more sources in the "Introduction" section and as a part of the discussion.
Our response: We have increased the number of references with the new information added in the text and also by extended the search until December 2023.
Revised text: See the revised text and references section.
Reviewer 2 Report
Comments and Suggestions for Authors
The purpose of the present study was to evaluate the prognostic factors that influence the outcome of periapical surgery.
if the authors speak about studies they have to cite more than two articles -Clinical studies on periapical surgery outcomes have reported success rates 32 ranging from 37% [1] to 91% [2].
Was the study registered in Prospero?
Why the search strategy was until 2021? Please extend until 2023 at least.
Please describe the algorithm- We used 65 algorithms and search strategies that can be reproduced by any researcher
Comments on the Quality of English Language
Moderate
Author Response
Responses to #reviewer 2:
- Concern of the reviewer: If the authors speak about studies they have to cite more than two articles -Clinical studies on periapical surgery outcomes have reported success rates 32 ranging from 37% [1] to 91% [2].
Our response: Thank you for your suggestion. We have cited more articles on periapical surgery outcomes.
Revised text: Line 45: …”ranging from 37% to 91% [5-10].”
2 . Concern of the reviewer: Was the study registered in Prospero?
Our response: Yes, the study was registered in Prospero with the registration number ID535642. We have added this information in the text.
Revised text: Lines 79-81: “Before starting the review the protocol was registered in PROSPERO with the registration number ID535642.”
- Concern of the reviewer: Why the search strategy was until 2021? Please extend until 2023 at least.
Our response: Thank you for your suggestion. We extended the search until December 2023.
Revised text: See the revised text, with the extended search and with the new references included. In addition, we have modified the figure 1.
- Concern of the reviewer: Please describe the algorithm- We used 65 algorithms and search strategies that can be reproduced by any researcher.
Our response: Thank you for your suggestion. The algorithms and search strategies have been described in the material and methods section.
Revised text: Lines 72-76: “The search strategy was: (((periapical surgery) OR (apical surgery) OR (endodontic surgery) OR (apical microsurgery) OR (periradicular surgery) OR (apicoectomy) OR (apicoectomy) OR (root-end resection)) AND ((healing) OR (prognosis factors) OR (guided tissue regeneration) OR (biomimetic membranes) NOT ((case report OR case reports OR in vitro OR experimental))).”
Lines 78-79: “We used algorithms and search strategies that can be reproduced by any researcher.”

Round 2
Reviewer 2 Report
Comments and Suggestions for Authors
The article was revised.